# Epidemiologic Impacts in Acute Infectious Disease Associated with Catastrophic Climate Events Related to Global Warming in the Northeast of Mexico

**DOI:** 10.3390/ijerph18094433

**Published:** 2021-04-22

**Authors:** Jesus Santos-Guzman, Francisco Gonzalez-Salazar, Gregorio Martínez-Ozuna, Victor Jimenez, Andrea Luviano, Daniel Palazuelos, Rubinia Iveth Fernandez-Flores, Mario Manzano-Camarillo, Esteban Picazzo-Palencia, Francisco Gasca-Sanchez, Gerardo Manuel Mejia-Velazquez

**Affiliations:** 1Tecnologico de Monterrey, Escuela de Medicina, Mexico, Ave. Morones Prieto #3000, Col. Los Doctores, Monterrey 64710, NL, Mexico; gmo@tec.mx (G.M.-O.); A01635398@itesm.mx (V.J.); andrealuvianog@gmail.com (A.L.); francisco.gasca@hotmail.com (F.G.-S.); 2Centro de Investigaciones Biomédica del Noreste, IMSS, Monterrey 64720, NL, Mexico; fgonz75@hotmail.com; 3Ciencias de la Salud, Universidad de Monterrey (UDEM), Monterrey 66238, NL, Mexico; 4Center for Research and Teaching in Economics (CIDE), Aguascalientes 20313, AG, Mexico; 5Brigham and Women’s Hospital, Harvard Medical School, Boston, MA 02115, USA; dpalazuelos@gmail.com; 6Tecnologico de Monterrey, Escuela de Ingeniería y Ciencias, Campus Monterrey, Monterrey 64849, NL, Mexico; rubiniafdz@gmail.com (R.I.F.-F.); mario.manzano@tec.mx (M.M.-C.); gmejia@tec.mx (G.M.M.-V.); 7Instituto de Investigaciones Sociales, UANL, Monterrey 64930, NL, Mexico; esteban.picazzopln@uanl.edu.mx; 8Departamento de Economia, Escuela de Negocios, Universidad de Monterrey (UDEM), San Pedro 66238, NL, Mexico

**Keywords:** climate change, global warming, water flood, border region, gastrointestinal diseases

## Abstract

Rising global temperatures and seawater temperatures have led to an increase in extreme weather patterns leading to droughts and floods. These natural phenomena, in turn, affect the supply of drinking water in some communities, which causes an increase in the prevalence of diseases related to the supply of drinking water. The objective of this work is to demonstrate the effects of global warming on human health in the population of Monterrey, Mexico after Hurricane Alex. We interpolated data using statistical downscaling of climate projection data for 2050 and 2080 and correlated it with disease occurrence. We found a remarkable rise in the incidence of transmissible infectious disease symptoms. Gastrointestinal symptoms predominated and were associated with drinking of contaminated water like tap water or water from communal mobile water tanks, probably because of the contamination of clean water, the disruption of water sanitation, and the inability to maintain home hygiene practices.

## 1. Introduction

Modern urban human life is characterized by demographic expansion, with increasing needs for food, housing, and energy, and consequently, by the generation of pollution and waste [1,2]. Current trends are worrisome, with pollution projected to rise over 50–70% within the next few years. This projection will have environmental consequences, such as a rise in global temperatures, a concomitant rise in seawater temperatures, and a related rise in extreme weather patterns [3]. 

Knutson associates the rise in ocean temperature with an increased number and intensity of hurricanes (2–11%), each with ever increasing destructive capabilities [4]. In Mexico, it is common to see the impact of these climate trends along the eastern border in the last 20 years. According to [5], the region reported several hurricanes and tropical storms (Figure 1).

These events represented important risks to an already vulnerable region; associated flooding and infrastructure damages disrupt economic development, worsen societal inequalities, and even jeopardize key components of the entire local ecosystem [6]. The increased risk of hurricanes relates not only to the ocean surface temperature, but also to the potential intensity variation factor (PIV). The calculation of PIV needs the following factors: the variation in net surface radiation, thermodynamic entropy, vorticity, and the mean wind speed at the surface. In the Atlantic Ocean, the PIV has increased by 10% in the last 40 years. Both the ocean surface temperature and the PIV are associated with a net increase in sea levels of radiative fluxes and decreasing tropopause temperatures [7].

In the last 20 years, this phenomenon has been occurring so quickly, and on such a vast scale, that the equilibrium threshold is often surpassed [8]. The effect on the ecosystem is often in the form of catastrophic natural events because of the profound changes in soil composition, hydrology, and the dynamics of water and air currents [9].

Epstein describes that most infectious diseases need a combination of an agent, a host, and a transmission environment; all of this elements are present in floods [10].

Liyanage et al. found increased dengue risk in Sri Lanka after a heavy rain (300 mm or more per week) and in Singapore after large cumulative precipitation [11].

The WHO linked malaria spread and flooding in Costa Rica during 1991, as well as in other countries, like Peru and other South American countries. During floods, basic human behavior changes, giving rise to issues with housing, overcrowding, decreased food and water safety, hygiene and more outdoors exposure to disease vectors [12]. In floods, waterborne disease follows drinking water contamination, disruption of sewages, and overflowing of infectious and toxic waste [13].

Health effects during floods depend on the functionality of the surviving public health infrastructure, the availability of healthcare services, and the rapidity, extent, and sustainability of the response after the disaster [14,15]. The effects of severe flooding described in the case of a 2010 monsoon flood in Pakistan represented an enormous amount of destruction and death: infrastructure was severely undermined with up to 40% of houses being destroyed, 95% of crops and 40% of livestock were lost, and even the land’s geography was deeply altered [16]. 

These cases illustrate how the modern urban lifestyle does not easily adapt to sudden disruptions caused by climate change. This is especially true in regions with rainy seasons where a large quantity of water can quickly be accumulated, but cannot be stored for later use, only to leave the area with a prolonged and disastrous period of drought. These floods not only cause infrastructural damages but also damage to the larger economy [17,18].

During the past few years, various epidemiological studies have shown the important correlation between floods and the rise of transmissible infectious disease [19,20,21]. Inundation of human settlements, infrastructure damages, and population displacement with flooding in Cambodia may reduce the clean water. The main diseases threatening the population in developing countries are diarrhea, dysentery, cholera, and typhoid [22].

It is generally recognized that climate change produced an extreme shift in climate, within seasons and years. Change in temperature patterns often leads to brisk accumulation of pluvial precipitation, increased hailstones, and frost. These phenomena can be interspersed with periods of drought and extreme heat. The warmer climate increases the proliferation of airborne and waterborne diseases.

Water availability connects to the agriculture sector, energy production, and waste and pollution production. These changes will alter the availability of water, its quality, agricultural/livestock productivity, and food security as well as financial security [23]. In short, climate change will promote the emergence and re-emergence of diseases, which may in turn lead to a reversal of the epidemiologic shift experienced by many developing nations.

Monterrey, Mexico, is the capital city of the state of Nuevo Leon. It is located in the northeast of Mexico, 215 km from the border with the United States of America. It is the second richest city in Mexico by the size of the local economy with a GDP of 90.837 million dollars. The Santa Catarina River, a river with very little water current, crosses the city of Monterrey from west to east when it is not the rainy season. However, during a heavy storm or a tropical hurricane that generates a large amount of rain, the riverbed becomes very dangerous and destructive, even deadly, due to the antecedents with Hurricane Gilberto (1988) and Hurricane Alex (2010). Hurricane Alex was type 2, at 175 km/h, coming from the southwest, producing 60 h of torrential rain in the inland city of Monterrey. The Santa Catarina River enlarged to 6500 m^2^/s of water when its capacity was overloaded and the city flooded. The National Water Commission estimated a water discharge of 616 mm/m^2^ during these events.

Monterrey is part of the Monterrey Metropolitan Area (MMA), the third largest metropolitan area in the country, although there is no single administrative entity for the metropolitan area. The municipalities that make up the MMA are as follows: Apodaca, Ciudad Benito Juárez, García, General Escobedo, Guadalupe, Santa Catarina, San Nicolás de los Garza, and San Pedro Garza García, Santiago, Cadereyta Jiménez and Salinas Victoria, its territorial extension is 451,300 km^2^, the average altitude of 500 m above sea level. Monterrey’s climate is semi-arid warm. The average annual rainfall is around 600 mm distributed mainly in the summer months, with September being the rainiest month of the year. In summer, the days are hot, with afternoons with temperatures close to 40 degrees and warm nights with temperatures close to 40 degrees. The hottest months of the year are July and August. In winter, the afternoons are pleasant, and the nights are cold.

López-Santos et al. measured the aridity index (AI) and laminar wind erosion trend in Durango, a city near Monterrey in the northeast of México, and estimated that in the next few years the climatic conditions of the area of study and its surroundings will deteriorate, increasing the risk for extreme conditions droughts and dust storms and promoting the heat island effect [24].

The objective of this paper is to highlight the impact of global warming effects on human health, as well as human preparedness on flood and city infrastructure quality, as illustrated through the increased incidence of infectious diseases in the Monterrey Metropolitan Area (MMA), located in the Northeastern border region of Mexico, caused by a hurricane.

## 2. Materials and Methods

We collected health-related clinical data from people that experienced the effects of flooding effects after Hurricane Alex, in the city of Monterrey, Nuevo Leon. The variables considered were faeco-orally transmitted diseases like diarrhea, dysentery, vomit, fever and gastric symptoms, other gastrointestinal symptoms, airborne diseases like fever and respiratory symptoms, sore throat, ocular pain, coughs, other respiratory symptoms, and cutaneous diseases like dermatitis and pruritus.

A random systematic method was used to include random study subjects who were then interviewed via telephone by trained study staff. We obtained climate and meteorological data from monitoring stations in the region and the Nuevo León State meteorological and air pollution database [25,26].

### 2.1. Climate Modeling

We used the scenarios of climate change estimated for Mexico and reported in the Climate Action Plan of the State of Nuevo León [27]. The A1B and A2 scenarios were developed for Mexico using a resolution of 0.5° × 0.5° for four periods of three decades each to estimate changes in maximum and minimum temperature and precipitation. The A1B Scenario describes the possible growth of a world with rapid economic growth, demographic growth with stabilization of the population growth by the second half of the century, and technology-based efficiency in human living, with balanced fossil and non-fossil energy sources. The A2 scenario describes self-reliance and preservation of local identities, unequal fertility rates but with variables increasing in the global population, unequal and variable productivity-dependent economic development, and a slow introduction to more efficient technologies [28].

The four periods covered were from 1980 to 2009, from 2010 to 2039, from 2040 to 2069, and from 2070 to 2099. The data of each scenario represents the probable changes of temperature and precipitation for the central decade and were named as the 1990s scenario, the 2020s scenario, the 2050s scenario, and the 2080s scenario, respectively. We downscaled the scenarios to a 5 km × 5 km grid to obtain higher resolution data for each three-decade period. We performed downscaling correlating scenario data of temperature and precipitation with historical data obtained from 118 climate meteorology stations within and surrounding the state of Nuevo León. We used the Lars—WG model to generate time series of minimum and maximum temperature as well as precipitation for the different periods of 1990s, 2020s, 2050s and 2080s for the two scenarios, A1B and A2. The ArcMap co-kriging statistical method was used to interpolate these variables. The variables used in the interpolation included the annual average readings of climate variables in each climate season, but also the specific altitude, exposure orientation, and distance to sea. The resulting interpolation layer was transformed to a RASTER format with a cell size of 5 km × 5 km. We prepared maps of temperature and precipitation for each of the three-decade periods considering the A1B and A2 scenarios.

We used the data of the high resolution grid to calculate the difference between the base data (period of the 1990s) and the estimated climatic change and pluvial precipitation in the next three-decade periods. The results show an expected temperature and pluvial precipitation change in the order of 0.51–1.5 °C and 6% respectively for the decades of 2010 to 2039. In the decades from 2040 to 2069 (the 2050s), the expected temperature change will be in the order of 1.01–1.50 °C with little change in pluvial precipitation. By the 2070 to 2099 decades (2080s), the expected temperature increase will be in the range of 2.01–2.50 °C. The pluvial precipitation expected to increase in the range of 0.1 and 2% in the A2 scenario and in the range of 6.1 a 10% in the A1B scenario [28]. For geodesic interpolation, we selected a 0.5° × 0.5° screen using the co-kriging geo-statistical method. Socio-demographic, climatic and seasonal variations, exposure vectors, and distance from the sea were incorporated in the calculations [29]. (Figure 2) In order to study the present and future effects of global warming, we selected several general circulation models of emission, applicable to the State of Nuevo Leon, México, based on socioeconomic factors.

### 2.2. Epidemiologic Health Effects Survey

This study is a retrospective observational descriptive study. After a recent flood in the metropolitan area of Monterrey, Nuevo Leon (N.L.), México, a study staff (mostly medical students) administered a telephone-based questionnaire to members of randomly selected households within the metropolitan area using a systematic residential telephone number selection method. To be eligible, respondents had to be 16 years of age or older and had to demonstrate knowledge of their own disease or the disease process of another family member and that were present during the period of June and July 2010. An expert committee validated this questionnaire. The telephonic survey questionnaire was administered within a 3-month time period after the height of the flooding, in order to diminish the recall bias. For the calculation of the incidence risk, the population of Monterrey in 2010 was 1,135,550 inhabitants (Source: INEGI. Censos y Encuestas Intercensales. http://datos.nl.gob.mx/n-l-poblacion-total-y-por-municipio/ (accessed on 23 March 2021).

## 3. Results

### 3.1. Climate and Temperture Modelation

In the modeling study, temperature shift and precipitation were calculated for both of the scenarios A1B and A2 (from 1960–1990 data), and the 2020s, 2050s, and 2080s three-decade periods. Figure 2 shows the general methodology followed. Figure 3 shows maps of the average precipitation and minimum and maximum temperatures in the state in the decades from 1960 to 1990. Figure 4 and Figure 5 show, respectively, maps of the expected changes in the values of the mean of the minimum and maximum temperatures for the decades of 2020, 2050, and 2080 [29]. 

### 3.2. Health Survey on Hurricane Alex (June 2010)

The results of the telephone survey initially included 2190 calls, 366 (16.7%) of which received no answer. Another 211 (11.6%) were excluded, either because the telephone line was not domestic, the person did not agree to answer, the questionnaire was incomplete, or the person that responded did not know the information of the household.

The sample included 1613 interviews available for analysis. Gender was distributed between 803 (49.8%) females, 704 (43.7%) males and 106 (6.6%) gender unspecified. Of them, 511 (31.7%) cases reported at least one symptom. The predominant symptoms were gastrointestinal (55.3%), followed by respiratory symptoms (38.0%) and dermatological symptoms (6.7%). There were 3.8 times more symptoms in July 2010, compared with the period of June 2010 (Table 1). Diarrhea and dysentery were significantly higher after Hurricane Alex (*p* = 0.029 y *p* = 0.045, respectively) (Figure 6). Respiratory diseases were significantly higher after Hurricane Alex (*p* = 0.029 y *p* = 0.045, respectively) (Figure 7). Of those with symptoms, 204 (33.7%) sought medical attention in June 2010, 118 (57.8%) in July 2010, and 51 (25%) during both months. Structural damage was reported by 447 (29.6%) of respondents.

## 4. Discussion

Based on our downscaled data of climate change modeling, the average temperatures might rise by 0.5–1 °C, and pluvial precipitation might rise by 6–10% for 90% of the State of Nuevo Leon within the next 10 years. With more floods expected, it might cause increased impact in human wellbeing and property structure. In addition, both waterborne and airborne transmissible diseases are expected. We found a remarkable rise in the incidence of transmissible infectious disease symptoms after Hurricane Alex. Gastrointestinal symptoms predominated and were associated with drinking tap water or water from communal mobile water tanks, probably because of the contamination of clean water, the disruption of water sanitation, and the inability to maintain home hygiene practices.

According to data from the Centro Nacional de Vigilancia Epidemiológica y Control de Enfermedades, the trend of rising temperatures and rainfall in the State of Nuevo Leon correlates positively with a rise in infections due to salmonellosis (50%), shigellosis (39%), typhoid fever (42%), and dengue (19%). After catastrophic events, an epidemic might be multifactorial and can include changes in human behavior, local climate, local biota, disturbance in normal hygiene, and the availability of necessary human resources. Common routes of transmission include the respiratory route, dermal route, blood-borne diseases, and the fecal-oral route. In an epidemic like malaria, the effects are seen with a lag time of 6–8 weeks after a flood [30]. In 2000, a flood in Mozambique [31] showed a 2–4 fold increase in the incidence of waterborne diseases. Our telephone survey data showed a similar 3.8 fold increase in the incidence of human infectious diseases after Hurricane Alex.

Respiratory diseases are also associated with climate change. In 2010, the US–Mexican border region reported 387.3 cases per 100,000 inhabitants [32]—an impressive 981% rise in the preceding 5-year period. One explanation of this large increase of cases are the improvements in diagnostic capabilities in local facilities, and to increases in pollution, pollen production, and climate change. Greenhouse gases can contribute to air pollution. For example, naturally produced and manmade CO_2_ pollution is associated with an increase in pollen production. CO_2_ can function as a plant fertilizer, thereby promoting its growth, which along with the increase in global temperatures might also promote an increase in pollen-producing plants and extend their growing period [33]. Early and more prolonged exposures to pollen and other air pollutants might explain the rising trends of asthma [34]. 

The chemical interaction of PM2.5 and ozone can produce secondary pollutants in the troposphere; these events can be triggered by sun exposure and high ambient temperatures. The climate change is producing altered wind patterns, extreme temperatures, and precipitation patterns [35,36]. Campbell-Lendrum et al. propose the term “polluting energy systems” and credits them with the increased burden of communicable disease linked to climate change and air pollution [37].

For example, in Monterrey, Mexico, Hurricane Alex-related flooding produced substantial damage: 6 deaths attributable to food, more than 15,000 internally displaced persons, a substantial increase in disease rates and job absenteeism of greater than 50%, and estimated economic losses of 1296 million dollars. The increasing risks associated with global climate change, including more frequent and serious meteorological conditions that increase the risk of severe flooding, obligate governments to produce better early warning systems, and new and better disaster readiness plans. This should include preparedness for more timely interventions that can prevent the early spread of diseases and more economic resources for the timely repair of damages.

Countries on either side of a shared border might experience different consequences from the same event. For example, while the US Centers for Disease Control have determined that flooding catastrophes do not normally increase the risk of infectious diseases in affected populations, on the Mexican side we determined that the incidence of such diseases is increased by at least 30%. The unequal outcomes are probably due to disparities in infrastructure, rapid respond programs, and resources, thereby reinforcing the idea that urban environments need improvements in order to meet these new challenges. If the expectation of flooding is at least every two years, more public works at all levels of the government (local, state, and federal levels) should work to address this issue. This may even include a formal city rebuilding, but should also involve rural sites and smaller cities. Infrastructure changes might include efforts from river embedding to changes in the ways to build houses. To do nothing is not an option as inaction may only lead to consequences of catastrophic dimensions. Risk management and adaptation are not a mere option, but rather an urgent need. A great amount of political will is necessary, and will hopefully be inspired by increased lobbying efforts with a growing general public awareness, education, and private sector participation.

Despite some progress of late, contingency countermeasures in Monterrey city will still need to improve in order to reduce the risks associated with flooding in different areas of the city. In 400 years of Monterrey’s history, there were 15 floods, 2 per century in the first three centuries, 7 in the twentieth century, and another 2 in the first decade of our current century. Floods are a relatively frequent event in the Monterrey metropolitan area and will become even more frequent as the climate changes. Public policy has a strong role to play in developing countermeasures that can address human health and human infrastructure damages and reduce the risk to the larger population. There is a need to improve the levies in the Santa Catarina riverbank, modernize the municipal sewage and waste disposal systems, and ensure the timelier provision of clean water and other needed services to the affected population. Each flood process should be studied with a root-cause analysis, deriving the implementation of preventive measures, including measurement of the efficiency and efficacy of the process. A limitation of the study was the climate model uncertainty, because the variable impact of hurricanes also depends on infrastructure/preparedness.

## 5. Conclusions

Global warming increases the risk of major hurricanes. Despise some progress in developing contingency countermeasures in Monterrey, Mexico, Hurricane Alex caused an increase in the incidence of symptoms of communicable infectious diseases, especially gastrointestinal symptoms associated with contamination of drinking water such as tap water or water from mobile communal water tanks. Disruption of water sanitation and inability to maintain hygiene practices in the home probably explain the main effects. There is an urgent need for improvements and rehabilitation of the riverbanks in the MMA and the urban drainage system.

## Figures and Tables

**Figure 1 ijerph-18-04433-f001:**
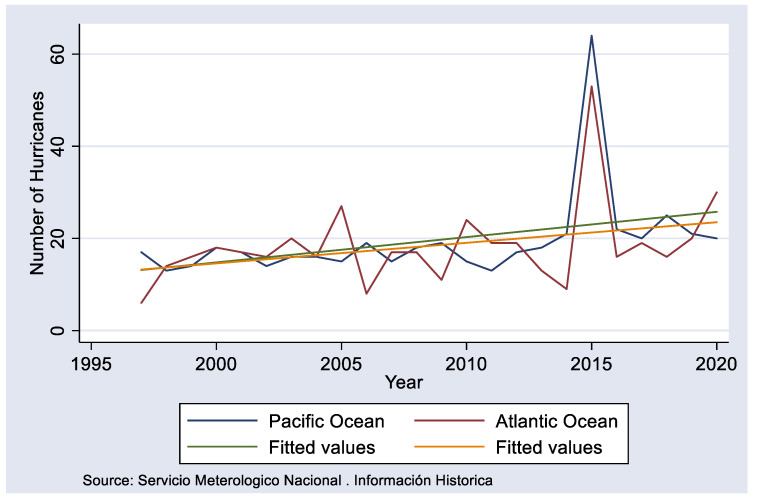
Hurricanes in the Mexican seas and shores.

**Figure 2 ijerph-18-04433-f002:**
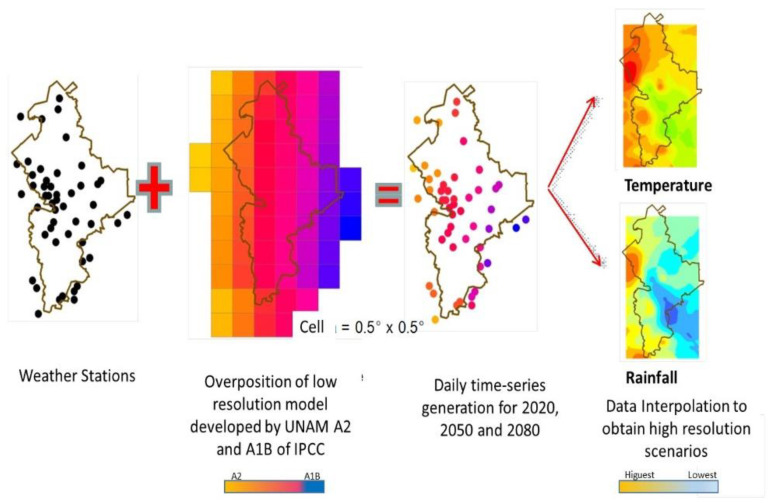
Modeling scheme for climate change in the State of Nuevo León, in 2020, 2050, and 2080. Model A2 in orange and AIB in blue.

**Figure 3 ijerph-18-04433-f003:**
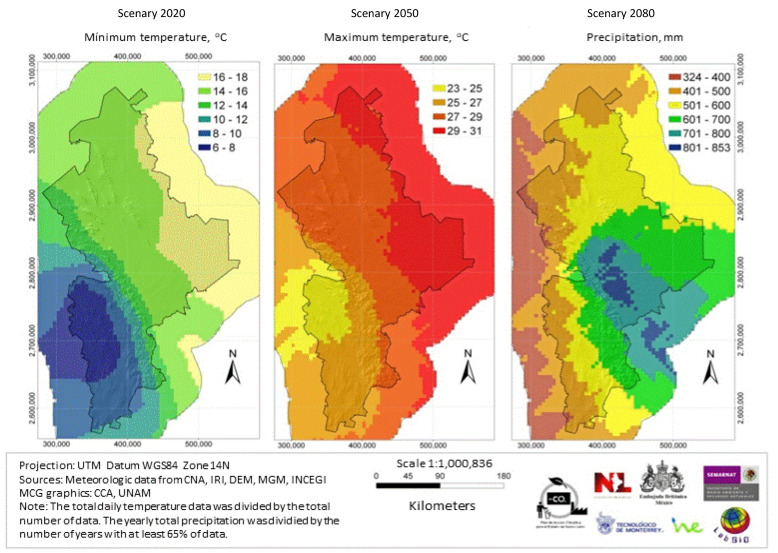
Mean precipitation and minimum and maximum temperature from 1960–1990.

**Figure 4 ijerph-18-04433-f004:**
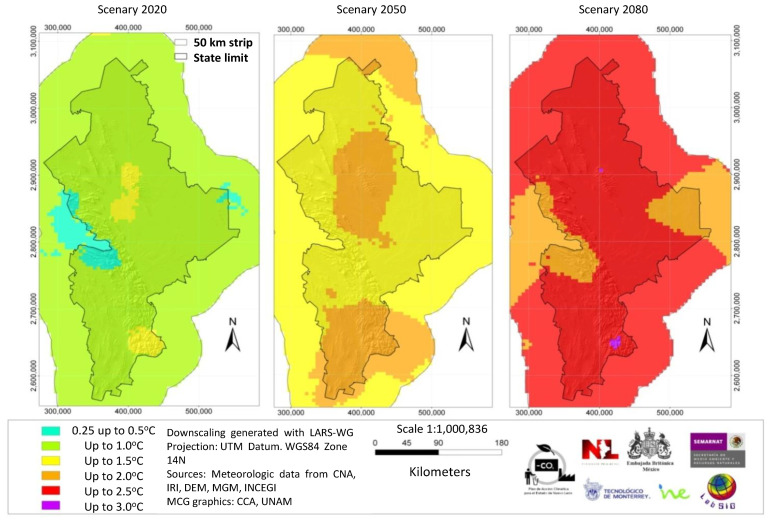
Changes in the mean minimal temperature in the A1B scenario for the 2020, 2050, and 2080 year periods, compared with the base period 1960–1990.

**Figure 5 ijerph-18-04433-f005:**
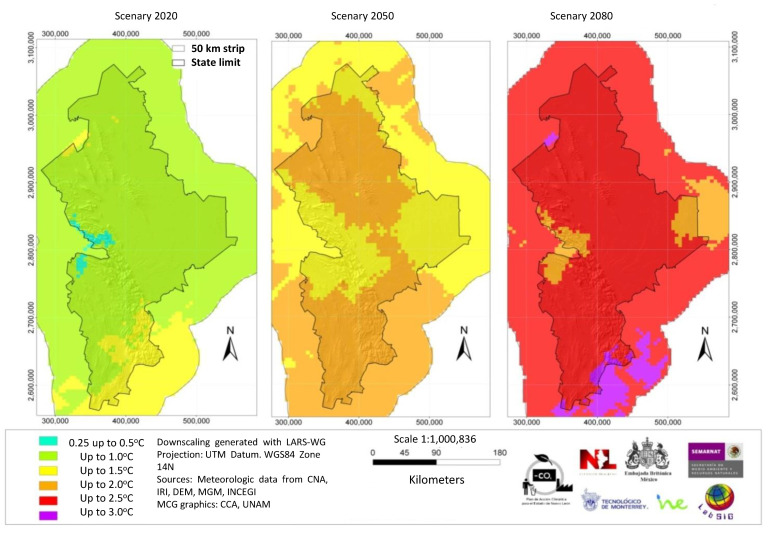
Changes in the mean maximum temperature in the A1B scenario for the 2020, 2050, and 2080 year periods, compared with the base period 1960–1990.

**Figure 6 ijerph-18-04433-f006:**
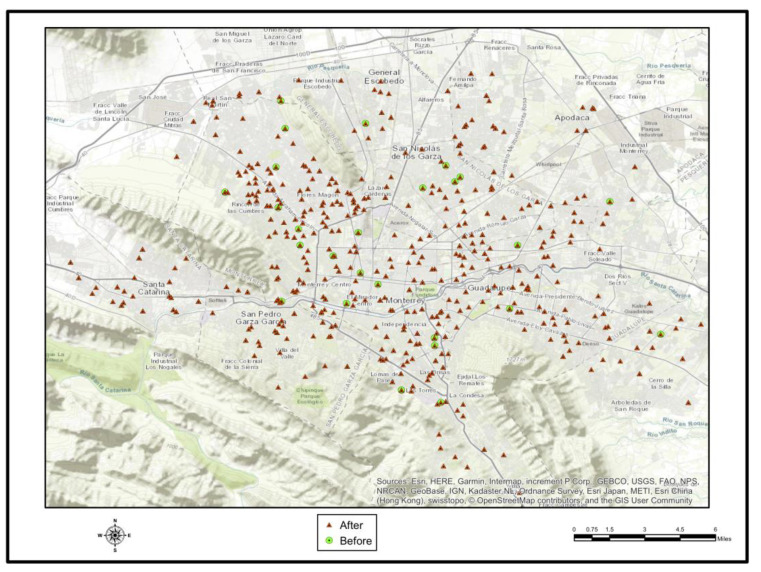
Showing the changes in the incidence of gastrointestinal diseases before and after Hurricane Alex.

**Figure 7 ijerph-18-04433-f007:**
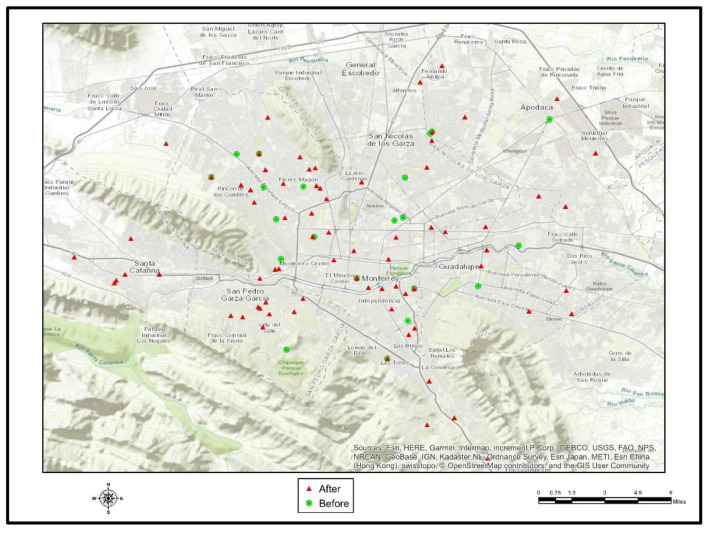
Showing the changes in the incidence of respiratory diseases before and after Hurricane Alex.

**Table 1 ijerph-18-04433-t001:** Incidence Rate per 100,000 inhabitants of Monterrey, during 2010, of symptoms before and after Hurricane Alex. (*n* = 1613 telephone surveys).

Symptoms	June (Before)	July (After)	Percentage of Change
Diarrhea	0.30	13.30	13.00
Dysentery	0.01	0.97	0.96
Vomit	0.33	0.00	−0.33
Fever and gastric symptoms	0.02	2.64	2.62
Other gastrointestinal symptoms	0.01	1.76	1.75
Fever and respiratory symptoms	0.04	2.99	2.95
Sore throat	0.15	4.67	4.52
Ocular pain	0.00	0.79	0.79
Cough	0.04	5.55	5.51
Other respiratory symptoms	0.00	1.59	1.59
Dermatitis	0.01	1.59	1.58
Pruritus	0.02	0.88	0.86
Total of symptoms	0.92	35.14	34.21

## Data Availability

The study did not report any data.

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
