# Peer review of "Epidemiologic Impacts in Acute Infectious Disease Associated with Catastrophic Climate Events Related to Global Warming in the Northeast of Mexico"

_ijerph, 2021, doi:10.3390/ijerph18094433_

Round 1
Reviewer 1 Report
Santos-Guzman et al. submitted the manuscript ”Epidemiologic Impact in Acute Infections Disease associated To Catastrophic Climate events related Global Warming in the northeast of Mexico'' for review. The paper investigated an interesting interdisciplinary topic between climate/weather and spread of disease. The subject is not new but the article presents an interesting case study on Hurricane Alex with convincing large surveys that illustrate the impact of hurricanes in addition to statistical downsizing results of climate projection that are interpolated to maps. The question of hurricane case increase and climate should be better explained, there a large literature on it. A better connection between the part may improve the quality of the paper and make it more smooth in its transition. Some important formatting should be done such as Figure captions on the bottom of the Figure. "." on the end of lines, no sub-section with Figure only (3.3) and more literature review on the link between climate and borne disease spread (Only flood in Mozambique [24) is mentioned. References formatting should be the same for all of them i.e. authors, date, title ...
Questions
#1 Would you suggest that 0.5°x0.5° grid resolution is sufficient for this study ? as it is relatively coarse climate projection (no dynamical downscaling)
#2 Regarding the link between climate change and pollution
Line 237 Greenhouse gases can contribute to air pollution
There is a strong assumption that the impact will be in the city, increasing urban canopy effect reducing circulation. That may contribute to your discussions.
#3 What further study should be done to improve the claim of the article ? More surveys ? Different hurricanes ? Different locations with a different infrastructure ?
General
Always lace Figure caption below the Figure and Table caption above (It is a standard convention)
Use indent just for the first sentence of a subsection and remove it in other places.
Place "." at end of sentence after the reference i.e. line 68
larger economy. [11,12]
-->
larger economy [11,12].
Try to get more recent climate data i.e. Fig 1 stopping in 2010 ?? 2020 data already available
Scenario projection of 2020 Figure 4 ? That is not a scenario in the current climate (currently 2021).
In your discussion Line 212-214, Try to connect the general GCM projection data that are general tendency with the hurricane case study smoothly with argument on the prediction of increase. You have to explain that the survey is only one case but with the confidence that it is representative of a typical hurricane situation, you also need to argue it.
Paragraphs on the limitation of the study should be include such as climate modl inertainty, the variable impact of hurricane depending of infrastructure/preparedness
Titles in the affiliation such as Professor or student should not be mentioned, simply the affiliation, add the country for each of them.
Abstract
Please, add a paragraph that summarized your methodology to achieve the research goal i.e. we interpolated used statistical downscaling of climate projection data and correlated it with disease occurrence ...
As this research also looks at climate scenarios 2020 (however not a scenario anymore) and 2050, you should also mention that part in the abstract. This is relatively important as it is a large part of the article.
Introduction
Line 72
Precise the conclusion of the study regarding the link between flooding causing inundation that damage infrastructure and contaminate clean water ... you should stress that point.
"Inundation of human settlements, infrastructure damage and population displacement with flooding in Cambodia may reduce clean water for drinking." in Davies et al., 2014
Line 74
, thereby leading to an accumulation of pluvial precipitation
-->
it generally recognized that climate change impact extreme. Not sure what the link between water availability and borne disease spread ... it is more extreme rainfall - flood --> water contamination --> disease spread. Water availability connects to the agriculture sector, energy production, etc ...
What is missing in your introduction is a state of the art knowledge on climate projection in the northeast of Mexico region. What the current knowledge, what the expected impact on rainfall ?
The link between borne disease spread and climate is already known in other parts of the world, I am unsure for that specific region but you should mention similar study on other areas and what their conclusion is.
- Materials and Methods
Please re-frame this part to give a clearer idea of what is data collection and what is method part in 2 distinctive sub-chapters.
Start by a sub paragraph 2.1 on Material and describe what was available on both clinical and climate data.
Make a 2.2. subchapter on Method
Move the data collection in the method followed by the Epidemiologic health effects survey part and Line 128-144 as well.
Keep the same order a-climate and then b-clinical or the other way.
Figure
Fig 1
Could you find a more recent Figure that shows the 2000-2020 tendency as data are now available.
Fig 2
Title the Figure as data processing as you used data extracted from a model (? did you made the climate run yourself ? It is unclear in this article)
You need to place the statistical downscaling on that chart otherwise it can be clearly understood, for instance the station location data (Daily time series) is obtained by merging climate scenario and station historical scenario. Then the statistical downscaling product is interpolated to obtain a gridded high resolution scenario ...
Fig 4
Increase the resolution of the Figure that x and y can be clearly seen.
Scenario 2020
-->
Scenario A1B, 2020
Scenario 2050
-->
Scenario A1B, 2050
Scenario 2080
-->
Scenario A1B, 2080
Fig 5
Scenario 2020
-->
Scenario A1B, 2020
Scenario 2050
-->
Scenario A1B, 2050
Scenario 2080
-->
Scenario A1B, 2080
Specific lines
Line 36
extreme weather patterns. [3]
-->
extreme weather patterns [3].
Line 38
tensity of hurricanes [2-11%],
-->
tensity of hurricanes (2-11%),
capabilities. [4]
-->
capabilities [4].
Line 40 Remove See
The region reported several hurricanes and tropical storms. [5] (See Fig. 1)
-->
According to [5], the region reported several hurricanes and tropical storms (Fig. 1).
Line 44 Remove the extra point there
Line 99 There is still uncertainty after the study as the link is non linear, it also depends on human preparedness on flood, city infrastructure quality etc ...
demonstrate the profound effects
-->
highlights the impact of global warming
Line 104 not useful for the paper, just state available data used and what method you used.
LIne 115 Add a sub-chapter title
Climate change modelling.
-->
2.1. Climate change modelling.
Add a 2.1. Data
Then add a 2.2. on your methods.
Then you can precise if you did the climate modelling or teh downscaling or interpolate the stations.
Line 128
For the calculation of SRES scenario, estimated for each meteorological station and a grid
of 0.5°x0.5°, we used the co-kriging geostatistical method.
-->
We used a co-kriging geostatistical method to produce 0.5°x0.5° gridded SRES scenario.
Line 141 Keep the same format for Figure reference all over the article i.e. Fig. xx.
calculations. [23] (See figure 2) In order to
-->
calculations [23] (Fig. 2). In order
Line 150
Remove that sentence (not useful)
Line 154
Not useful, it is simply in by a research paper ...
Line 162
Here start with an indent as it is a new sub-chapter.
Line 163
during the basal measurement
->
historical model
Line 172, 175 and 179
--> Add a . at the end of the caption
Line 182
3.2. Health survey
-->
3.2. Health survey on Alex hurricane case (June 2010)
Line 200 Figures can not solely be a sub-chapter. You need to incorporate the Figures in the text 3.2 and remove the 3.3. section. Strat by citing the Figure, then add the Figure and make it analysis.
3.3. Figures, Tables and Schemes
Line 209
Based on our climate change modeling
-->
Clarify her that you use climate modelling data that you downscale ...
Line 269 Better to not frame it as personal
Despite some progress of late, contingency countermeasures in our city
-->
Despite some progress of late, contingency countermeasures in Monterrey city.
Line 271
we have had
-->
there were
Line 281
In Monterrey, Mexico, Hurricane
-->
For example, in Monterrey, Mexico, Hurricane
Author Response
RESPONSE TO EDITOR 1
(RESPONSES ARE IN RED INK)
Open Review
(x) I would not like to sign my review report
( ) I would like to sign my review report
English language and style
( ) Extensive editing of English language and style required
( ) Moderate English changes required
( ) English language and style are fine/minor spell check required
(x) I don't feel qualified to judge about the English language and style
|
Yes |
Can be improved |
Must be improved |
Not applicable |
|
|
Does the introduction provide sufficient background and include all relevant references? |
( ) |
(x) |
( ) |
( ) |
|
Is the research design appropriate? |
( ) |
(x) |
( ) |
( ) |
|
Are the methods adequately described? |
( ) |
( ) |
(x) |
( ) |
|
Are the results clearly presented? |
( ) |
( ) |
(x) |
( ) |
|
Are the conclusions supported by the results? |
( ) |
(x) |
( ) |
( ) |
Comments and Suggestions for Authors
Santos-Guzman et al. submitted the manuscript ”Epidemiologic Impact in Acute Infections Disease associated To Catastrophic Climate events related Global Warming in the northeast of Mexico'' for review. The paper investigated an interesting interdisciplinary topic between climate/weather and spread of disease. The subject is not new but the article presents an interesting case study on Hurricane Alex with convincing large surveys that illustrate the impact of hurricanes in addition to statistical downsizing results of climate projection that are interpolated to maps.
The question of hurricane case increase and climate should be better explained, there a large literature on it.
A better discussion with bibliography was added, on climate and pollution, climate and floods, climate and disease
A better connection between the part may improve the quality of the paper and make it more smooth in its transition. Some important formatting should be done such as Figure captions on the bottom of the Figure. "." on the end of lines, no sub-section with Figure only (3.3) and more literature review on the link between climate and borne disease spread (Only flood in Mozambique [24) is mentioned. All of the above done
References formatting should be the same for all of them i.e. authors, date, title ... Done
Questions
#1 Would you suggest that 0.5°x0.5° grid resolution is sufficient for this study ? as it is relatively coarse climate projection (no dynamical downscaling)
The Scenarios of climate change in temperature and precipitation were estimated for Mexico in a previous study using a resolution of 0.5o x 0.5o for the three-decade periods from 1980 to 2009, from 2010 to 2039, from 2040 to 2069 and from 2070 to 2099. Each of these four periods were named as the scenarios for the central decade, i.e., 1990s scenario, 2020s scenario, 2050s scenario and 2080s scenario, respectively. The data of these estimated scenarios were downscaled to a grid of 5 km x 5 km correlating with historical data of climate meteorology stations in the state and surrounding the state of Nuevo León using the Lars-WG model to generate times series of minimum and maximum temperature as well as precipitation for the different periods of 2020s, 2050s and 2080s for both scenarios, A1B and A2. The ArcMap cokriging statistical method was used to interpolate these variables. The resulting interpolation layer was transformed to a RASTER format with a cell size of 5 km x 5 km. These high resolution grid data were used in this study.
We clarify these in the manuscript.
#2 Regarding the link between climate change and pollution
Line 237 Greenhouse gases can contribute to air pollution The text was changed in discussion
The chemical interaction of PM2.5 and ozone to produce secondary pollutants in the troposphere is promoted by sun exposure and higher ambient temperatures. The climate change is producing altered wind patterns, extreme temperatures and precipitation patterns.
Kinney, P.L. Interactions of Climate Change, Air Pollution, and Human Health. Curr Envir Health Rpt 5, 179–186 (2018). https://doi.org/10.1007/s40572-018-0188-x
Orru, H., Ebi, K.L. & Forsberg, B. The Interplay of Climate Change and Air Pollution on Health. Curr Envir Health Rpt 4, 504–513 (2017). https://doi.org/10.1007/s40572-017-0168-6
Campbell-Lendrum et al. propose the term polluting energy systems, were much of the incremented burden of communicable disease is linked to climate change and air pollution
Diarmid Campbell-Lendrum and Annette Prüss-Ustün. Climate change, air pollution and noncommunicable diseases. Bull World Health Organ. 2019 Feb 1; 97(2): 160–161
For example, in
There is a strong assumption that the impact will be in the city, increasing urban canopy effect reducing circulation. That may contribute to your discussions.
The text was changed in the introduction
Epstein describe that most infectious disease need and agent, a host and transmission environment, all present in floods.
- R. Epstein, “Climate change and emerging infectious diseases,” Microbes and Infection, vol. 3, no. 9, pp. 747–754, 2001.
Liyanage et al found increased dengue risk in Sri Lanka after a heavy rain (300mm or more per week) and in Singapore after large cumulative precipitation.
Liyanage, P. et al. Int. J. Environ. Res. Public Health 13, 1087 (2016).
Liyanage, P. et al. Int. J. Environ. Res. Public Health 13, 1087 (2016).
WHO, linked Malaria spread and flooding in Costa Rica during 1991, Peru and other South American countries. During floods basic human behavior changes, including housing, overcrowding, decreased food and water safety, hygiene and more outdoors exposure to disease vectors.
WHO. Humanitarian Health Action. Flooding and communicable diseases fact sheet. (Last consulted February 28, 2021) https://www.who.int/hac/techguidance/ems/flood_cds/en/#:~:text=Floods%20can%20potentially%20increase%20the,fever%2C%20and%20West%20Nile%20Fever
In floods, waterborne disease follows drinking water contamination, disruption of sewages and overflowing of infectious and toxic waste.
Lu Gao, Ying Zhang, Guoyong Ding, Qiyong Liu, and Baofa Jiang. Identifying Flood-Related Infectious Diseases in Anhui Province, China: A Spatial and Temporal Analysis. The American Journal of Tropical Medicine and Hygiene. 2016;94(4) 741-749. https://www.ncbi.nlm.nih.gov/pmc/articles/PMC4824213/#!po=67.2414
Health effects during floods depends on the functionality of the surviving public health infrastructure, the availability of healthcare services, and the rapidity, extent, and sustainability of the response after the disaster.
Ivers, Louise Ca,b,c; Ryan, Edward Tb,c,d Infectious diseases of severe weather-related and flood-related natural disasters, Current Opinion in Infectious Diseases: October 2006 - Volume 19 - Issue 5 - p 408-414 doi: 10.1097/01.qco.0000244044.85393.9e
Azar Shokri, Sadaf Sabzevari, Seyed Ahmad Hashemi. Impacts of flood on health of Iranian population: Infectious diseases with an emphasis on parasitic infections. Parasite Epidemiology and Control. 2020;9: e00144. https://doi.org/10.1016/j.parepi.2020.e00144.
#3 What further study should be done to improve the claim of the article ? More surveys ? Different hurricanes ? Different locations with a different infrastructure ? The text was changed in discussion
There is a need to improve the levies in the Santa Catarina river´s bank, modernizing the municipal sewage and waste disposal systems, and the timelier provision of clean water and other needed services to the affected population. Each flood process should be studies with a root-cause analysis, deriving to the implementation of preventive measures, including measurement of the efficiency and efficacy of the process. A limitation of the study was the climate model uncertainty, because the variable impact of hurricane also depends on infrastructure/preparedness.
General
Always lace Figure caption below the Figure and Table caption above (It is a standard convention)
The text was changed
Use indent just for the first sentence of a subsection and remove it in other places.
Place "." at end of sentence after the reference i.e. line 68
larger economy. [11,12]
-->
larger economy [11,12]. The text was changed
Try to get more recent climate data i.e. Fig 1 stopping in 2010 ?? 2020 data already available
The text was changed
se hizo una nueva Tabla 1 con datos de Servicio Metereologico Nacional de 1997 al 2020
Scenario projection of 2020 Figure 4 ? That is not a scenario in the current climate (currently 2021).
We consider the four three-decades periods prepared by the Mexican government to prepare climate action plans for A1b and A2 scenarios. These data were used to prepare high resolution scenarios for the state of Nuevo León. We used the same terminology The three decades from 1980 to 2009 is named the 1990 scenario, the decades from 2010 to 2039 is named as the 2020 scenario, the decades from 2040 to 2069 is the 2050 scenario, and the decades from 2070 to 2099 is the 2080 scenario. The three decades from 2010 to 2039 covers the expected changes in temperature and precipitation expected in the 2021 year.
We clarify these in the manuscript.
In your discussion Line 212-214, Try to connect the general GCM projection data that are general tendency with the hurricane case study smoothly with argument on the prediction of increase. You have to explain that the survey is only one case but with the confidence that it is representative of a typical hurricane situation, you also need to argue it.
The text was changed in the introduction
Epstein describe that most infectious disease need and agent, a host and transmission environment, all present in floods.
- R. Epstein, “Climate change and emerging infectious diseases,” Microbes and Infection, vol. 3, no. 9, pp. 747–754, 2001.
Liyanage et al found increased dengue risk in Sri Lanka after a heavy rain (300mm or more per week) and in Singapore after large cumulative precipitation.
Liyanage, P. et al. Int. J. Environ. Res. Public Health 13, 1087 (2016).
Liyanage, P. et al. Int. J. Environ. Res. Public Health 13, 1087 (2016).
WHO, linked Malaria spread and flooding in Costa Rica during 1991, Peru and other South American countries. During floods basic human behavior changes, including housing, overcrowding, decreased food and water safety, hygiene and more outdoors exposure to disease vectors.
WHO. Humanitarian Health Action. Flooding and communicable diseases fact sheet. (Last consulted February 28, 2021) https://www.who.int/hac/techguidance/ems/flood_cds/en/#:~:text=Floods%20can%20potentially%20increase%20the,fever%2C%20and%20West%20Nile%20Fever
In floods, waterborne disease follows drinking water contamination, disruption of sewages and overflowing of infectious and toxic waste.
Lu Gao, Ying Zhang, Guoyong Ding, Qiyong Liu, and Baofa Jiang. Identifying Flood-Related Infectious Diseases in Anhui Province, China: A Spatial and Temporal Analysis. The American Journal of Tropical Medicine and Hygiene. 2016;94(4) 741-749. https://www.ncbi.nlm.nih.gov/pmc/articles/PMC4824213/#!po=67.2414
Health effects during floods depends on the functionality of the surviving public health infrastructure, the availability of healthcare services, and the rapidity, extent, and sustainability of the response after the disaster.
Ivers, Louise Ca,b,c; Ryan, Edward Tb,c,d Infectious diseases of severe weather-related and flood-related natural disasters, Current Opinion in Infectious Diseases: October 2006 - Volume 19 - Issue 5 - p 408-414 doi: 10.1097/01.qco.0000244044.85393.9e
Azar Shokri, Sadaf Sabzevari, Seyed Ahmad Hashemi. Impacts of flood on health of Iranian population: Infectious diseases with an emphasis on parasitic infections. Parasite Epidemiology and Control. 2020;9: e00144. https://doi.org/10.1016/j.parepi.2020.e00144.
Paragraphs on the limitation of the study should be include such as climate modl inertainty, the variable impact of hurricane depending of infrastructure/preparedness
The text was changed at the end of discussion
A limitation of the study was the climate model incertainty, because the variable impact of hurricane also depends on infrastructure/preparedness
Titles in the affiliation such as Professor or student should not be mentioned, simply the affiliation, add the country for each of them.
The text was changed affiliation not needed were erased, keeping only affiliation. Also one affiliation (6) was added and the change of name Andrea Josefina Luviiano was changed by Andrea Luviano.
Jesus Santos-Guzman1,*, Francisco Gonzalez-Salazar3 Gregorio Martínez-Ozuna.1Victor Jimenez1, Andrea Luviano1,6, Daniel Palazuelos2, Rubinia Iveth Fernandez-Flores4, Mario Manzano-Camarillo4, Esteban Picazo5, Francisco Gasca-Sanchez1 and Gerardo Manuel Mejia-Velazquez4
|
|
1 Profesor of the Escuela de Medicina, Tecnológico de Monterrey Monterrey.
2 Pre-graduate student of the Escuela de Medicina, Tecnológico de Monterrey.
2 Chief Strategist, Co-founder, Compañeros en Salud – Mexico; Instructor, Harvard Medical School
3 Centro de Investigaciones Biomédica del Noreste, IMSS y Ciencias de la Salud, Universidad de Monterrey (UDEM)
5 Post-graduate Student Tecnológico de Monterrey, Campus Monterrey.
4 Professor of the School of Engineering, Tecnológico de Monterrey, Campus Monterrey.
5 Instituto de Investigaciones Sociales, UANL
6 Center for Research and Teaching in Economics (CIDE), Aguascalientes, México
Abstract
Please, add a paragraph that summarized your methodology to achieve the research goal i.e. we interpolated used statistical downscaling of climate projection data and correlated it with disease occurrence ...
The text was changed
The objective of this work is to demonstrate the effects of global warming on human health in the population of the Monterrey, Mexico after the Hurricane Alex. We interpolated data using statistical downscaling of climate projection data for 2050 and 2080 and correlated it with disease occurrence.
As this research also looks at climate scenarios 2020 (however not a scenario anymore) and 2050, you should also mention that part in the abstract. This is relatively important as it is a large part of the article.
The scenario for the the decades from 2010 to 2039 is known as the 2020 scenario, The year 2021 is within this period.
We clarify these in the manuscript.
Introduction
Line 72
Precise the conclusion of the study regarding the link between flooding causing inundation that damage infrastructure and contaminate clean water ... you should stress that point.
"Inundation of human settlements, infrastructure damage and population displacement with flooding in Cambodia may reduce clean water for drinking." in Davies et al., 2014
. Inundation of human settlements, infrastructures damage and population displacement with flooding in Cambodia may reduce the clean water [16].
Line 74
, thereby leading to an accumulation of pluvial precipitation
-->
it generally recognized that climate change impact extreme. Not sure what the link between water availability and borne disease spread ... it is more extreme rainfall - flood --> water contamination --> disease spread. Water availability connects to the agriculture sector, energy production, etc ...
The text was changed in the introduction
……diseases threatening the population in developing countries are diarrhea, dysentery, cholera and typhoid [16].
It is generally recognized that climate change impact extreme shift in climate within seasons and years. Change in temperature patterns often leads to brisk accumulation of pluvial precipitation, increased hailstones, frost, interspersed with periods of drought and extreme heat. The warmer climate increases the proliferation of airborne and waterborne diseases. Prolonged changes in temperature can promote a shift in climate within the seasons and along the years, thereby leading to an accumulation of pluvial precipitation, increased hailstones, frost, with periods of drought intercalated throughout.
Water availability connects to the agriculture sector, energy production, waste and pollution production.
These changes will alter …….
What is missing in your introduction is a state of the art knowledge on climate projection in the northeast of Mexico region. What the current knowledge, what the expected impact on rainfall ?
Text added to the introduction López-Santos et. al. measured the aridity index (AI) and laminar wind erosion trend in the northeast of México and estimate than in the next few years the climatic conditions of the area of study and its surroundings will deteriorate incrementing the risk for extreme conditions droughts and dust storms, promoting the heat island effect. [lopez-Santos]
Lopez-Santos, A.; Pinto Espinoza, J.; Ramirez Lopez, E. M. y Martinez Prado, M. A..Modeling the potential impact of climate change in northern Mexico using two environmental indicators. Atmósfera [online]. 2013, vol.26, n.4, pp.479-498. ISSN 0187-6236.
The link between borne disease spread and climate is already known in other parts of the world, I am unsure for that specific region but you should mention similar study on other areas and what their conclusion is.
The text was changed in the introduction
- Materials and Methods
Please re-frame this part to give a clearer idea of what is data collection and what is method part in 2 distinctive sub-chapters.
Start by a sub paragraph 2.1 on Material and describe what was available on both clinical and climate data.
Make a 2.2. subchapter on Method
Move the data collection in the method followed by the Epidemiologic health effects survey part and Line 128-144 as well.
Keep the same order a-climate and then b-clinical or the other way.
Figure
Fig 1
Could you find a more recent Figure that shows the 2000-2020 tendency as data are now available.
The figure was changed in the introduction
Fig 2
Title the Figure as data processing as you used data extracted from a model (? did you made the climate run yourself ? It is unclear in this article)
We clarify these in the manuscript.
You need to place the statistical downscaling on that chart otherwise it can be clearly understood, for instance the station location data (Daily time series) is obtained by merging climate scenario and station historical scenario. Then the statistical downscaling product is interpolated to obtain a gridded high resolution scenario ... The text was changed in Material and methods also Figure caption were modified
We used a co-kriging geostatistical method to produce 0.5°x0.5° gridded SRES scenario and data were statistically donwscaled. For the calculation of SRES scenario, estimated for each meteorological station and a grid of 0.5°x0.5°, we used the co-kriging geostatistical method. The variables used in the interpolation included: annual average readings of climate
Figure 4. Changes in the mean minimal temperature in the downscaled A1B scenario for the 2020, 2050 and 2080 year-periods, compared with the base period 1960-1990.
Figure 5. Changes in the mean maximum temperature in the donwscaled A1B scenario for the 2020, 2050, and 2080 year-periods, compared with the base period 1960-1990
Fig 4
Increase the resolution of the Figure that x and y can be clearly seen.
Scenario 2020
-->
Scenario A1B, 2020 The figure was changed
Scenario 2050
-->
Scenario A1B, 2050 The figure was changed
Scenario 2080
-->
Scenario A1B, 2080 The figure was changed
Fig 5
Scenario 2020
-->
Scenario A1B, 2020 The figure was changed
Scenario 2050
-->
Scenario A1B, 2050 The figure was changed
Scenario 2080
-->
Scenario A1B, 2080 The figure was changed
Specific lines
Line 36
extreme weather patterns. [3]
-->
extreme weather patterns [3]. The figure was changed
Line 38
tensity of hurricanes [2-11%],
-->
tensity of hurricanes (2-11%), The text was changed in introduction
capabilities. [4]
-->
capabilities [4]. The figure was changed
Line 40 Remove See
The region reported several hurricanes and tropical storms. [5] (See Fig. 1)
-->
According to [5], the region reported several hurricanes and tropical storms (Fig. 1). The text was changed in introduction
Line 44 Remove the extra point there The text was changed
Line 99 There is still uncertainty after the study as the link is non linear, it also depends on human preparedness on flood, city infrastructure quality etc ... The text was changed
The objective of this paper highlights the impact of global warming effects of global warming on human health, as well as human preparedness on flood and city infrastructure quality, as illustrated through the increased incidence of infectious diseases in the Monterrey Metropolitan Area (MMA), located in the Northeastern border region of Mexico, caused by a hurricane.
demonstrate the profound effects
-->
highlights the impact of global warming The text was changed in introduction
Line 104 not useful for the paper, just state available data used and what method you used. The text was changed in Materials and methods
This paper is descriptive, prospective, analytic, and approved by a research committee. We collected health related clinical data from people that experienced the effects of flooding effects after Hurricane Alex, in the city of Monterrey, Nuevo Leon. The variables considered were faeco-orally transmitted diseases like diarrhea, dysentery, vomit, fever and gastric symptoms, other gastrointestinal symptoms, airborne disease like fever and respiratory symptoms, sore throat, ocular pain, coughs, other respiratory symptoms and cutaneous diseases like dermatitis and pruritus.
LIne 115 Add a sub-chapter title
Climate change modelling.
-->
2.1. Climate change modelling. The text was changed in Materials and methods
The objective of this paper highlights the impact of global warming effects of global warming on human health, as well as human preparedness on flood and city infrastructure quality, as illustrated through the increased incidence of infectious diseases in the Monterrey Metropolitan Area (MMA), located in the Northeastern border region of Mexico, caused by a hurricane.
- Materials and Methods
This paper is descriptive, prospective, analytic, and approved by a research committee.
Add a 2.1. Data
Then add a 2.2. on your methods. The text was changed in Materials and methods
2.2 Epidemiologic health effects survey
This study is a retrospective observational descriptive study. After
Then you can precise if you did the climate modelling or teh downscaling or interpolate the stations.
We climate modelling was donwscalled and was declared in the paper
Line 128
For the calculation of SRES scenario, estimated for each meteorological station and a grid
of 0.5°x0.5°, we used the co-kriging geostatistical method.
-->
We used a co-kriging geostatistical method to produce 0.5°x0.5° gridded SRES scenario.
The text was changed in Materials and methods
We used a co-kriging geostatistical method to produce 0.5°x0.5° gridded SRES scenario. For the calculation of SRES scenario, estimated for each meteorological station and a grid of 0.5°x0.5°, we used the co-kriging geostatistical method. The variables used in the interpolation included: annual average readings of climate variables in each climate season, but also, specific altitude, exposure orientation
Line 141 Keep the same format for Figure reference all over the article i.e. Fig. xx.
calculations. [23] (See figure 2) In order to
-->
calculations [23] (Fig. 2). In order The text was changed in Materials and methods
For geodesic interpolation, we selected a 0.5°x0.5° screen using the co-kriging geo-statistical method. Socio-demographic, climatic and seasonal variations, exposure vectors and distance from the sea were incorporated in the calculations [23]. (Figure 2) In order to study
Line 150
Remove that sentence (not useful) The text was changed in Materials and methods
Line 154
Not useful, it is simply in by a research paper ... The text was changed in Materials and methods
2.2 Epidemiologic health effects survey
This study is a retrospective observational descriptive study. After a recent flood in the metropolitan area of Monterrey, N.L., México, a study staff (mostly medical students) administered a telephone-based questionnaire to members of randomly selected households within the metropolitan area using a systematic residential telephone number selection method. The study staff received capacitation and supervision by their professor. To be eligible, respondents had to be 16 years of age or older and had to demonstrate knowledge of their own disease or the disease process of other family member and that were present during the period of June and July 2010. An expert committee validated this questionnaire. The telephonic survey questionnaire was administered within a 3-month time period after the height of the flooding, in order to diminish the recall bias. For the calculation of the incidence risk, a population of Monterrey in 2010 was 1,135,550 inhabitants (Source: INEGI. Censos y Encuestas Intercensales. http://datos.nl.gob.mx/n-l-poblacion-total-y-por-municipio/)
Line 162
Here start with an indent as it is a new sub-chapter. ... The text was changed
Line 163
during the basal measurement
->
historical model The text was changed in results
In the modeling study, temperature shift and precipitation wer alculated for both of the scenarios A1B and A2 (from 1960-1990 data), and the 2020s, 2050s and 2080s periods. Figure 2
Line 172, 175 and 179
--> Add a . at the end of the caption The text was changed in results
Line 182
3.2. Health survey
-->
3.2. Health survey on Alex hurricane case (June 2010) The text was changed in results
3.2. Health survey on Alex Huricae case (June 2010)
The results of the telephone survey initially included 2,190 calls, 366 (16.7%) of which received no answer. Another
Line 200 Figures can not solely be a sub-chapter. You need to incorporate the Figures in the text 3.2 and remove the 3.3. section. Strat by citing the Figure, then add the Figure and make it analysis. The text was changed in results
3.3. Figures, Tables and Schemes
3.3. Figures, Tables and Schemes
Line 209
Based on our climate change modeling
-->
Clarify her that you use climate modelling data that you downscale ... The text was changed in discussion
- Discussion
Based on our downscalled data of climate change modeling, the average temperatures might rise by 0.5oC -1oC, and pluvial precipitation
Line 269 Better to not frame it as personal
Despite some progress of late, contingency countermeasures in our city
-->
Despite some progress of late, contingency countermeasures in Monterrey city. The text was changed in discussion
Despite some progress of late, contingency countermeasures in our Monterrey city will still need to improve in order to reduce the risks associated with flooding in different areas of the city. In 400 years of Monterrey’s history, we have had there were 15 floods, two per century in the first three centuries, seven in the twentieth Century,
Line 271
we have had
-->
there were The text was changed in discussion
Line 281
In Monterrey, Mexico, Hurricane
-->
For example, in Monterrey, Mexico, Hurricane The text was changed in discussion
…..pollen producing plants and extend their growing period. [26] Early and more prolonged exposures to pollen and other air pollutants might explain the rising trends of asthma [27]. For example, in Monterrey, Mexico, Hurricane Alex related flooding in Monterrey city produced substantial damage: 6 deaths attributable to food, more than 15,000 internally displaced

Reviewer 2 Report
The objective of this study is to examine the effects of global warming on infectious diseases in the population of the Monterrey, Mexico. To accomplish the principal objective of the study, the authors compared patterns of infectious diseases before and after the landfall of the Hurricane Alex. There is no description of the hurricane in the text and the study assumed that the event occurred due to global climate change without providing any evidence. In fact, the connection between climate change and formation of hurricane is still being studied without any definite conclusion. Ideally, health impacts of any hurricane should be studied by comparing the incidence of infectious diseases before and after the event. In my opinion, the study wrongly analyzed the impact by people who became ill in both periods (before and after) (Table 1). In this table, "symptoms" should be replaced by "illness" and the "absoule numbers" should be replaced by "rate of incidence," which should be calculated. Other comments are as follows:
- Given the objective of the study, the sub-section "Climate change modelling" (line number 115) is irrelevant (also with the title of the manuscript) and needs to be deleted.
- Figure 1 is not needed - it does not prove anything.
- More information is needed for questionnaire survey
- In several places, the meaning of senteces is not clear (e.g., line numbers 37-38 and 111). The manuscript needs editing.
Author Response
RESPONSES TO EDITOR 2
(RESPONSES ARE IN RED INK)
Open Review
(x) I would not like to sign my review report
( ) I would like to sign my review report
English language and style
( ) Extensive editing of English language and style required
(x) Moderate English changes required
( ) English language and style are fine/minor spell check required
( ) I don't feel qualified to judge about the English language and style
|
Yes |
Can be improved |
Must be improved |
Not applicable |
|
|
Does the introduction provide sufficient background and include all relevant references? |
(x) |
( ) |
( ) |
( ) |
|
Is the research design appropriate? |
( ) |
( ) |
(x) |
( ) |
|
Are the methods adequately described? |
( ) |
( ) |
(x) |
( ) |
|
Are the results clearly presented? |
( ) |
( ) |
(x) |
( ) |
|
Are the conclusions supported by the results? |
(x) |
( ) |
( ) |
( ) |
Comments and Suggestions for Authors
The objective of this study is to examine the effects of global warming on infectious diseases in the population of the Monterrey, Mexico. To accomplish the principal objective of the study, the authors compared patterns of infectious diseases before and after the landfall of the Hurricane Alex.
There is no description of the hurricane in the text and the study assumed that the event occurred due to global climate change without providing any evidence.
The Hurican Alex was type 2, with 175km/h, coming from the southwest, producing 60 h of torrential raining in the inland city of Monterrery, the Santa Catarina River enlarge to 6,500 m2/sec of water when its capacity was overloaded and the city flooded. The National Water Comission estimated a water discharche of 616 mm/m2 during these events
In fact, the connection between climate change and formation of hurricane is still being studied without any definite conclusion. Ideally, health impacts of any hurricane should be studied by comparing the incidence of infectious diseases before and after the event.
Text was added to associate climate change, floods and disease presence:
Epstein describe that most infectious disease need and agent, a host and transmission environment, all present in floods.
- R. Epstein, “Climate change and emerging infectious diseases,” Microbes and Infection, vol. 3, no. 9, pp. 747–754, 2001.
Liyanage et al found increased dengue risk in Sri Lanka after a heavy rain (300mm or more per week) and in Singapore after large cumulative precipitation.
Liyanage, P. et al. Int. J. Environ. Res. Public Health 13, 1087 (2016).
Liyanage, P. et al. Int. J. Environ. Res. Public Health 13, 1087 (2016).
WHO, linked Malaria spread and flooding in Costa Rica during 1991, Peru and other South American countries. During floods basic human behavior changes, including housing, overcrowding, decreased food and water safety, hygiene and more outdoors exposure to disease vectors.
WHO. Humanitarian Health Action. Flooding and communicable diseases fact sheet. (Last consulted February 28, 2021) https://www.who.int/hac/techguidance/ems/flood_cds/en/#:~:text=Floods%20can%20potentially%20increase%20the,fever%2C%20and%20West%20Nile%20Fever
In floods, waterborne disease follows drinking water contamination, disruption of sewages and overflowing of infectious and toxic waste.
Lu Gao, Ying Zhang, Guoyong Ding, Qiyong Liu, and Baofa Jiang. Identifying Flood-Related Infectious Diseases in Anhui Province, China: A Spatial and Temporal Analysis. The American Journal of Tropical Medicine and Hygiene. 2016;94(4) 741-749. https://www.ncbi.nlm.nih.gov/pmc/articles/PMC4824213/#!po=67.2414
Health effects during floods depends on the functionality of the surviving public health infrastructure, the availability of healthcare services, and the rapidity, extent, and sustainability of the response after the disaster.
Ivers, Louise Ca,b,c; Ryan, Edward Tb,c,d Infectious diseases of severe weather-related and flood-related natural disasters, Current Opinion in Infectious Diseases: October 2006 - Volume 19 - Issue 5 - p 408-414 doi: 10.1097/01.qco.0000244044.85393.9e
Azar Shokri, Sadaf Sabzevari, Seyed Ahmad Hashemi. Impacts of flood on health of Iranian population: Infectious diseases with an emphasis on parasitic infections. Parasite Epidemiology and Control. 2020;9: e00144. https://doi.org/10.1016/j.parepi.2020.e00144.
In my opinion, the study wrongly analyzed the impact by people who became ill in both periods (before and after) (Table 1). In this table, "symptoms" should be replaced by "illness" and the "absoule numbers" should be replaced by "rate of incidence," which should be calculated. Done in table
|
Table 1. Incidence Rate per 100, 000 inhabitants of Monterrey, durning 2010 of Symptoms before and after Alex Hurricane |
|||||
|
(n=1,613 telephone surveys) |
|||||
|
Illness |
June (before) |
July (after) |
Percentage of change |
||
|
Diarrhea |
0.30 |
13.30 |
13.00 |
||
|
Disentery |
0.01 |
0.97 |
0.96 |
||
|
Vomit |
0.33 |
0.00 |
-0.33 |
||
|
Fever and gastric symptoms |
0.02 |
2.64 |
2.62 |
||
|
Other gastrointestinal. symptoms |
0.01 |
1.76 |
1.75 |
||
|
Fever and respiratory symptoms |
0.04 |
2.99 |
2.95 |
||
|
Sore throath |
0.15 |
4.67 |
4.52 |
||
|
Ocular pain |
0.00 |
0.79 |
0.79 |
||
|
Cought |
0.04 |
5.55 |
5.51 |
||
|
Other respiratory symptoms |
0.00 |
1.59 |
1.59 |
||
|
Dermatitis |
0.01 |
1.59 |
1.58 |
||
|
Pruritus |
0.02 |
0.88 |
0.86 |
||
|
Total of symptoms |
0.92 |
35.14 |
34.21 |
||
Other comments are as follows:
- Given the objective of the study, the sub-section "Climate change modelling" (line number 115) is irrelevant (also with the title of the manuscript) and needs to be deleted. The text was changed in material and methods
A random systematic method was used to include study subjects were randomly, and then interviewed via telephone by trained study staff. We obtained climate and meteorological data from monitoring stations in the region and the Nuevo León State meteorological and air pollution database [18, 19].
2.1 Climate change modelling.
We used the results of a general circulation model in order to predict climatic changes and trends for the State of N.L. From the SRES scenarios, we selected A1B y A2 for the years representing the decades of 2020s, 2050s and 2080s [20].
- Figure 1 is not needed - it does not prove anything. The figure was changed in the introduction.
It might be relevant because it shows the growing tendency of huricanes in Mexico Data was added to include all country, not just Monterrey and from 1997 to 2020.
- More information is needed for questionnaire survey
Epidemiologic health effects survey
- This study is a retrospective observational descriptive study. After a recent flood in the metropolitan area of Monterrey, N.L., México, a study staff (mostly medical students) administered a telephone-based questionnaire to members of randomly selected households within the metropolitan area using a systematic residential telephone number selection method. The study staff received capacitation and supervision by their professor. To be eligible, respondents had to be 16 years of age or older and had to demonstrate knowledge of their own disease or the disease process of other family member and that were present during the period of June and July 2010. The questionnaire was validated by an expert committee. The telephonic survey questionnaire was administered within a 3-month time period after the height of the flooding, in order to diminish the recall bias. For the calculation of the incidence risk a polullation of Monterrey in 2010 was use the population of 1,135,550 inhabitants (Source: INEGI. Censos y Encuestas Intercensales. http://datos.nl.gob.mx/n-l-poblacion-total-y-por-municipio/)
- In several places, the meaning of senteces is not clear (e.g., line numbers 37-38 and 111). The manuscript needs editing.
These line an others were reviewed
Submission Date
15 February 2021
Date of this review
22 Feb 2021 18:31:44
